# InDel and SCoT Markers for Genetic Diversity Analysis in a Citrus Collection from the Western Caucasus

**DOI:** 10.3390/ijms24098276

**Published:** 2023-05-05

**Authors:** Raisa Kulyan, Lidiia Samarina, Ruset Shkhalakhova, Alexandr Kuleshov, Yulia Ukhatova, Olga Antonova, Natalia Koninskaya, Alexandra Matskiv, Valentina Malyarovskaya, Alexey Ryndin

**Affiliations:** 1Federal Research Centre the Subtropical Scientific Centre of the Russian Academy of Sciences, 354002 Sochi, Russia; raisa.kulyan22@gmail.com (R.K.); shhalahova1995@mail.ru (R.S.); mister.alexandr.ru@gmail.com (A.K.); natakoninskaya@mail.ru (N.K.); matskiv_a@mail.ru (A.M.); malyarovskaya@yandex.ru (V.M.); ryndin@subtropras.ru (A.R.); 2“Sirius University of Science and Technology”, Olimpiyskiy Ave. b.1, 354340 Sirius, Kransnodar Region, Russia; sci_secretary@vir.nw.ru (Y.U.); olgaant326@mail.ru (O.A.); 3Federal Research Center N. I. Vavilov All-Russian Institute of Plant Genetic Resources (VIR), 190000 Saint Petersburg, Russia

**Keywords:** citrus, cold tolerance, germplasm collections, SCoT markers, InDel markers, genetic diversity, phylogenetic relationship, population structure

## Abstract

Citrus collections from extreme growing regions can be an important source of tolerant germplasms for the breeding of cold-tolerant varieties. However, the efficient utilization of these germplasms requires their genetic background information. Thus, efficient marker systems are necessary for the characterization and identification of valuable accessions. In this study, the efficiency of 36 SCoT markers and 60 InDel markers were evaluated as part of the broad citrus collection of the Western Caucasus. The interspecific and intraspecific genetic diversity and genetic structures were analyzed for 172 accessions, including 31 species and sets of the locally derived cultivars. Single markers, such as SCoT18 (0.84), SCoT20 (0.93), SCoT23 (0.87), SCoT31 (0.88), SCoT36 (0.87) и LG 1-4 (0.94), LG 4-3 (0.86), LG 7-11 (0.98), and LG 8-10 (0.83), showed a high discriminating power, indicating the good applicability of these markers to assess intraspecific diversity of the genus *Citrus*. Overall, SCoT markers showed a higher level of polymorphism than InDel markers. According to analysis of population structure, SCoT and InDel markers showed K = 9 and K = 5 genetic clusters, respectively. The lowest levels of genetic admixtures and diversity were observed among the locally derived satsumas and lemons. The highest level of genetic admixtures was observed in the lime group. Phylogenetic relationships indicated a high level of interspecific genetic diversity but a low level of intraspecific diversity in locally derived satsumas and lemons. The results provide new insight into the origin of citrus germplasms and their distribution in colder regions. Furthermore, they are important for implementing conservation measures, controlling genetic erosion, developing breeding strategies, and improving breeding efficiency.

## 1. Introduction

Commercially important cultivars of the genus *Citrus* L. are the most produced fruits in the world with over 153 million tons of production in 2018 [1,2]. 

The Western Caucasus is one of the most northerly Citrus growing areas in the world where the winter temperatures can fall below minus 5 °C and the absolute minimum temperature is minus 11 °C [3]. It makes this region unique for the cultivation and breeding of cold-tolerant citrus genotypes. Citrus germplasm was introduced here in 1902 from the Mediterranean, USA, Turkey, Iran, and Japan. As a result of the conventional breeding, dozens of locally adapted cultivars of lemon, mandarin, pomelo, etc. have been developed over the past 80 years. Currently, this citrus collection consists of more than 200 accessions maintained in the FRC SSC RAS, which were phenotypically characterized over decades [4]. The sets of locally derived accessions are of unknown origin, some of them forming from asexual reproduction [5,6] and others from free pollination and controlled hybridization. Among them are a broad set of frost-tolerant lemon and mandarin genotypes [4]. To understand the mechanisms of the domestication of Citrus germplasm in this extreme environment, the genetic origin and relationships among accessions have to be elaborated. In addition, the genetic background information of this germplasm is necessary for its efficient utilization. However, a high level of polyembryony is a serious breeding constraint for many citrus varieties, thus it is necessary to find efficient DNA markers with high discriminative power for intraspecific fingerprinting.

Different DNA markers (AFLP, IRAP, SSR, CAPS, SCoT, InDel) were evaluated to characterize the genetic diversity in the Citrus germplasm collections worldwide [7,8,9,10,11,12,13]. These studies indicated that each marker type has its advantages and disadvantages. Thus, the combination of different markers can be an efficient tool to characterize the genetic diversity in collections. Among the different DNA markers, the insertion–deletion (InDel) markers were recently developed based on the full genome sequencing of satsuma mandarin to identify hybrid embryos [9]. The advantage of these markers is that they can distinguish heterozygous and homozygous genotypes using a simple and inexpensive method of agarose gel electrophoresis. However, the efficiency of these markers for other Citrus species has still not been sufficiently evaluated [10]. Additionally, the start codon-targeted (SCoT) markers are based on polymorphisms in the short, conserved region of plant genes surrounding the ATG translation initiation codon and are codominant due to insertion–deletion mutations [14]. Since the region flanking the ATG start codon is highly conserved in all plant species, it was predicted that the SCoT method would be useful for generating DNA markers in diverse plant species [15].

The objective of this study was to evaluate the efficiency of InDel and SCoT markers for the interspecific and intraspecific analysis of different citrus species and to evaluate the relationships and genetic background of one of the northernmost citrus collections. The results provide new insight into the citrus germplasm’s origin and distribution in colder regions, and they are important for implementing conservation measures, controlling genetic erosion, developing breeding strategies, and improving breeding efficiency.

## 2. Results

### 2.1. Efficiency of SCoT and InDel Primers for Genetic Diversity Analysis of the Citrus Germplasm Collection

Out of 36 SCoT primers, 24 primers showed low amplification quality with weak or fuzzy bands and were therefore removed from the analysis. The remaining 12 SCoT primers showed reproducible results with clear polymorphisms and resolution within the citrus genotypes (SCoT2, SCoT6, SCoT9, SCoT12, SCoT13, SCoT14, SCoT18, SCoT20, SCoT21, SCoT23, SCoT31, SCoT36) (Appendix A). With the 12 SCoTs, a total of 322 bands were detected in 172 genotypes, ranging from 19 (for SCoT36) to 35 (for SCoT12) bands with the mean diversity index H = 0.43 (Table 1). An average PIC = 0.34 was detected with the highest value being 0.39 for SCoT2 and the lowest value being 0.32 for SCoT18. The mean discriminating power D = 0.52 was detected with the highest value ranging from 0.84 to 0.93 for SCoT18, SCoT20, SCoT23, SCoT31, and SCoT36.

Out of 60 InDel primers, 51 primers showed a low amplification quality with weak or fuzzy bands and were therefore removed from the analysis. The remaining 9 InDel primers showed reproducible results with clear polymorphisms and resolution within *Citrus* genotypes (LG 1-4, LG 2-6, LG3-14, LG4-2, LG 4-3, LG 5-20, LG 7-11, LG 8-10, LG 9-2) (Appendix A).

For the 9 InDels, a total of 40 bands were detected in 172 genotypes, ranging from 2 (for LG 2-6, LG3-14) to 11 (for LG 7-11) with the mean diversity index H = 0.38 (Table 2). An average PIC = 0.41 was detected, with the highest value being 0.46 for LG 2-6 and the lowest value being 0.37 for LG 8-10. The mean D = 0.64 was observed with the highest value ranging from 0.83 to 0.98 for LG 1-4, LG 4-3, LG 7-11, and LG 8-10.

### 2.2. Genetic Structure of the Citrus Germplasm Collection Based on SCoT and InDel Polymorphisms

Following STRUCTURE HARVESTER analysis using the 12 SCoT marker data, the 172 accessions were grouped into nine genetic clusters (K = 9) (Figure 1). The genetic structure based on SCoT markers showed clear clustering of citrus species. Overall, each genetic cluster consisted of several species of the genus *Citrus*. The highest level of homogeneity was observed in the clusters 1, 2, 5, 6, and 7. Interestingly, 40 cultivars of satsuma mandarin (*C. × aurantium* var. *unshiu*) were divided into two clusters: cluster 1 (mostly combined new locally derived breeding forms) and cluster 3 (mostly combined old locally derived cultivars). Among them, a set of the genotypes with genetic admixtures was identified, for example, Hybrid 98-9, Clone 31, Sentyabrskii, Olimpiyskii2014, Kodorskii, Kelasurskii, Clone#33, Hybrid 99-4, Clone#22, Izeki Wase, Sochinskii23, Slava Vavilova, or Pioneer80. These genotypes can be an important source of diversity and will be useful for further breeding programs to increase the genetic diversity of mandarins in colder regions. All true lemons (*C. × limon* var. *limon*), along with the ancestral species *C. medica* and its hybrids, comprised the green cluster 2 and all oranges comprised the purple cluster 5. Several genotypes with genetic admixtures of about 10–50% were observed in each of these clusters.

The highest level of genetic admixtures was observed in Cluster 4 (yellow color). This cluster combined *C. maxima* accessions and its hybrid species, citron, bitter orange, and hybrids of *C. × junos*. The remaining 8 clusters mostly consisted of sweet limes and sour limes with their respective ancestral species (brown color). Most lime accessions contained a high percentage of genetic admixtures of the other citrus groups.

The InDel markers showed less clear genetic structure as compared to the SCoT markers. As a result, several secondary species and interspecific hybrids were combined into the same clusters (Figure 2). In particular, red cluster-combined accessions belonged to several species: sweet oranges (*C. × aurantium* var. *sinensis*) and limes (*C. × latifolia* var. *latifolia* and *C. × limon* var. *limetta*, *C. aurantifolia*). Green cluster-combined *Fortunella* spp. Accessions combined grapefruits (*C. × aurantium* var. *paradisi*) and its hybrids. Finally, the last two clusters consisted of the lemons (*C. × limon* var. *limon*) (blue cluster) and mandarins (*C. × aurantium* var. *unshiu*) (yellow cluster) with a low level of genetic admixtures. Additionally, two varieties of bitter orange (*C. × aurantium* L. var. *aurantium*) and *C. ×junos* joined the mandarin group.

To summarize, both SCoT and InDel marker data showed efficient discrimination of satsuma mandarins and lemons, which were grouped into separate clusters and showed a homogeneous genetic structure with a low level of admixtures. Additionally, SCoT markers allowed for the separation of orange, kumquat, and lime accessions, showing a greater level of polymorphisms as compared to InDel markers. However, both marker types showed a low level of genetic admixtures in *C. × aurantium* var. *unshiu*, *C. × aurantium* var. *sinensis*, and *C. × limon* var. *limon*, except for the few accessions showing 5 to 20% admixtures.

### 2.3. Phylogenetic Analysis of the Citrus Germplasm Collection Based on SCoT and InDel Polymorphisms

The results of the neighbor joining analysis based on the combined SCoT and InDel data resulted in three main branches divided into seven sub-branches. These are presented on a phylogenetic tree (Figure 3). The first branch divided into two sub-branches. Sub-branch 1.1 combined 33 accessions, 31 of which belonged to the lemon group, including old local cultivars such as “Odishi”, “Novoafonskii”, “Beskolyuchii”, “Udarnik”, “Hybrid3252”, etc. Their probable closely related hybrids of citron and mandarin joined this group. Sub-branch 1.2 combined 25 accessions of sweet lime varieties, as well as hybrids and the ancestral species of *C. medica* and *C. maxima*, indicating their close relationship.

The most abundant branch 2 combined three sub-branches of mandarins and oranges. Sub-branch 2.1 combined 38 accessions. Particularly, 12 accessions of sweet and bitter oranges and 14 accessions of satsuma mandarin of foreign and local origin joined this group. In addition, several accessions of pomelos and grapefruits were closely grouped with them. Sub-branch 2.2 combined 26 accessions of *C. × aurantium* var. *unshiu* of local breeding with non-presentable genetic distances among them. Sub-branch 2.3 combined 18 accessions of *C. reticulata* and its hybrids and *C. × leiocarpa*.

Finally, Branch 3 showed the greatest genetic distances among accessions and was divided into two sub-branches: Sub-branch 3.1 contained accessions of *Fortunella* spp., *Poncirus* spp., and wild relatives. Sub-branch 3.2 grouped several lime accessions of *C. × limon* var. *limetta*, *C. × latifolia* var. *latifolia*, and *C. aurantifolia*.

To summarize the results of the neighbor joining analysis, the combination of SCoT and InDel markers resulted in plausible positioning of the species on the tree. The clear separation of citrus species and the low level of genetic diversity in the locally derived cultivars, especially in satsuma mandarins and lemons («Millenium1», «Shirokolistnyy», «Yubileinii», «Iveriya», «Sochinskii23» «Slava Vavilova», «Pioneer80», «Sakharny», «Millenium 2», «Оcho Wase», «Kolkhidskii», «Kelasurskii», «Sentyabrskii», «Krasnodarskii83», «Olimpiyskii2014», «Georgievskii», «Krupnoplodny», etc. and «Odishi», «Novoafonskii», «Beskolyuchii», «Kuznera», «Novogruzinskii», «Maykopskii», «Krupnoplodny», «Udarnik», «Gonio», «Dioskuriya», «Gagrinskii» etc.) were revealed. The highest level of genetic diversity was observed in Branch 3 (wild relatives), Sub-branch 2.3 (mandarins), and Sub-branch 1.2 (limes). These results show the necessity of interspecific hybridization to increase genetic diversity in the local lemon and satsuma genotypes.

## 3. Discussion

The collection of the FRC SSC RAS is located on the Black Sea coast of the Western Caucasus and is represented by the genetic and ecological–geographical diversity of wild, introduced citrus species and cultivars, including locally derived cultivars. This is a unique border region for the cultivation and breeding of new cold-tolerant citrus varieties. One of the most important problems associated with the efficient utilization of citrus genetic resources is the insufficient use of modern germplasm characterization tools [16,17]. In this study, we analyzed the efficiency of SCoT and InDel markers to assess the genetic structure and diversity of the local citrus collection of 172 accessions.

### 3.1. Efficiency of SCoT and InDel Markers for Genetic Diversity Analysis of the Citrus Germplasm Collection

The genome of some citrus species has only recently been published, particularly the genome of the satsuma mandarin [18,19,20]. Based on the satsuma genome, InDel markers were recently developed to identify closely related accessions in the satsuma collection [9,10]. The satsumas are the most cold hardy mandarin varieties, and a high level of polyembryony is a serious biological constraint. Thus, it is necessary to find efficient markers with high discriminative power for intraspecific fingerprinting. In this study, we also selected SCoT markers, which can also be useful for QTL mapping due to advantages such as low cost and ease of use, as well as a connection with the sites of transcription initiation. Despite the wide use of SCoT markers, studies on citrus are insufficient and usually conducted on a small number of accessions [12,13,21].

For the first time, we have evaluated the efficiency of SCoT and InDel markers on a wide range of citrus species and cultivars.

Polymorphism information content (PIC) and discriminating power (D) characterize the degree of capabilities, efficiency, and potential of marker systems [22]. The PIC corresponds to its ability to detect polymorphisms among the individuals in a population [23]. The maximum PIC value for dominant markers is 0.5 [24,25,26,27,28,29]. Our results from the dominant SCoT markers showed a mean PIC = 0.3. For co-dominant markers, the efficient PIC is believed to be 0.5–1.0 [30]. Our results from the co-dominant InDel markers showed a mean PIC = 0.4, indicating a poor ability to indicate polymorphisms in the broad range of citrus species. The low PIC values for InDel markers were also reported in several other studies [31,32].

The D represents the probability that two randomly chosen individuals have different allelic patterns, and thus are distinguishable from one another [30]. The mean D value was 0.52 for SCoT and 0.64 for InDel markers. Several markers showed especially high values of D, namely SCoT18 (D = 0.84), SCoT20 (D = 0.93), SCoT23 (D = 0.87), SCoT31 (D = 0.88), SCoT36 (D = 0.87) и LG 1-4 (D = 0.94), LG 4-3 (D = 0.86), LG 7-11 (D = 0.98), and LG 8-10 (D = 0.83), indicating a good applicability of these markers to assess intraspecific diversity in the genus *Citrus*.

### 3.2. Genetic Structure of the Citrus Germplasm Collection Based on SCoT and InDel Polymorphisms

Following a STRUCTURE HARVESTER, the SCoT and InDel markers showed K = 9 and K = 5 genetic clusters, respectively. A different number of K can be explained by different discriminating powers and different genome target regions amplified by InDel and SCoT markers [33].

We suggested that InDel markers would be efficient to reveal the detailed genetic structure of the satsuma collection and identify hybrids because these markers were designed for satsuma species [10]. However, a low level of genetic admixtures was observed in satsuma clusters in particular, which is consistent with our previous study conducted with SSR markers [34]. Another study reported that InDel markers showed a low intraspecific diversity in several citrus species [35]. In general, a discrepancy between the morphological diversity and genetic homogeneity of cultivars within a species is common [34,35,36,37] since most varieties of secondary species originated from a clonal breeding and the fixation of somatic mutations [19,38,39]. Apomixis in mandarins, oranges, and lemons is also a possible reason for the low intraspecific genetic diversity.

Based on the selected SCoT and InDel markers, the genetic admixtures in several genotypes of mandarin («Sakharny», «Sochinskii23», «Clone33», «Hybrid 99-4», «Slava Vavilova»), lemon («Moskovskii», «Udarnik», «Gonio»), sweet orange («Washington Navel», «Washington Navel dwart», «Sukhumskii», «Hamlin»), and pomelo («Yubileinyy», «Asahikan», «Metelyova», «Grushevidnyy», «Sambokan») were clearly identified. These genotypes will be useful for further controlled hybridization to increase intraspecific genetic diversity in these species. Additionally, there was a significant introgression in the lime group with a genetic admixture of micranta and citron, which is consistent with other studies [12,40]. The admixtures of lemon, tangerine, lime, and orange were observed in pomelo and citron varieties and trifoliate orange. This indicates that sexual reproduction more frequently occurs in mono-embryonic wild relatives and ancestral citrus species rather than in secondary hybrid species, which is consistent with other studies [6,19,33].

### 3.3. Phylogenetic Analysis of the Citrus Germplasm Collection Based on SCoT and InDel Polymorphisms

The phylogenetic relationships between citrus and its relatives were successfully identified based on the selected SCoT and InDel markers. Accessions of wild *Poncirus* spp. And *Fortunella* spp., as well as the ancestral species *C. maxima*, *C. medica*, *C. reticulata*, and *C. micrantha* showed a comparable genetic distance to other citrus species.

The first branch showed an ancestral relationship to *C. limon* var. *limon* and *C. × limon* var. *limetta*. The origin of lemons and limes is complex and has different phylogenetic roots. It is suggested that *C. lemon* var. *limon* originated from the hybridization of *C. × aurantium* (*C. maxima* × *C. reticulata*) × *C. medica* or was obtained through a mutation of the original hybrid [19]. Sweet lime *C. × limon* var. *limetta* is believed to have the same parents as *C. limon* var. *limon*, but is thought to be derived from an independent reticulation event [40]. Our results are consistent with these data and demonstrate the close relations of *C. limon* var. *limon*, *C. × limon* var. *limetta*, *C. maxima*, and *C. medica*. Additionally, sour lime accessions (*C. × latifolia* var. *latifolia*) indicated close relationships with *C. aurantifolia* and *C. micrantha* and as such, they were placed in the third branch of the phylogenetic tree. *C.× latifolia* var. *latifolia* was derived from the pollination of the haploid ovule of *C. limon* var. *limon* by the diploid gamete of *C. aurantifolia*. In turn, *C. aurantifolia* was derived from the natural hybridization of *C. micrantha* as a female parent and *C. medica* as a male parent [40,41,42]. However, despite the fact that *C. × limon* var. *limetta* mainly contains the genetic contributions of *C. maxima*, *C. reticulata*, and *C. medica*, several accessions of sweet lime were located in the same branch with *C. × latifolia* var. *latifolia.* This could happen due to the low number of lime accessions in our dataset.

It is interesting that almost all locally derived cultivars of satsuma mandarin grouped into a separate sub-branch of the phylogenetic tree. It is assumed that satsuma mandarin (*C. × aurantium* var. *unshiu*) is a Japanese breeding line of the mandarin *C. reticulata* var. *austera*, which originated from the crossing of different species of mandarin and a number of independent somatic mutations [41,43]. Sweet orange (*C. × aurantium* var. *sinensis*) is placed nearby satsuma mandarins and it is considered a hybrid of the two ancestral species of *C. reticulata* var. *austera* and *C. maxima* [44]. Hybrid species *C. × aurantium* var. *clementina* and *C. × junos* were placed closer to *C. × aurantium* var. *unshiu* on the phylogenetic tree. These species presumably originated from crosses between *C. × aurantium* var. *sinensis* × *C. reticulata* var. *austera* and *C. reticulata* var. *austera* × *C. cavaleriei*, respectively [44]. Our results are consistent with other phylogenetic studies where *C. maxima*, *C. × aurantium* var. *sinensis*, *C. cavaleriei*, and *C. reticulata* var. *austera* acted as the common genetic contributors to mandarin-related species [44,45]. According to the neighbor joining analysis, *C. × yuko* and *C. × leiocarpa* grouped in the same sub-branch, which is consistent with the assumption that *C. × yuko* is a cross between *C. × kinokuni* and *C. × leiocarpa* [46]. Thus, differentiation between the gene pools of the original species *C. maxima*, *C. medica*, *C. reticulata*, and *C. micrantha* acted as a structuring factor of the analyzed edible citrus germplasm. Secondary species and modern varieties, in general, were intermediate between the main taxa, confirming their hybrid status.

## 4. Materials and Methods

### 4.1. The Plant Material and DNA Extraction

The plant material of 172 Citrus genotypes, including 31 species, 43 local cultivars, and 22 locally derived hybrids, was obtained from the germplasm bank of the FRC SSC RAS (Appendix A). The fresh samples of young leaves were obtained from a healthy plant of each genotype. The plants were 5–30 years old and were maintained in the field and greenhouse collection. One sample per genotype was taken as a biological replicate. An amount of 70–100 mg of each sample was ground and mixed into a liquid nitrogen. DNA extraction was performed using CTAB protocol [47]. DNA quality was checked by agarose gel electrophoresis and spectrophotometrically using BioDrop μLite (Biodrop, Cambridge, UK). All samples were diluted to 20 ng μL^−1^ and stored at −20 °C.

### 4.2. PCR Analysis and Visualization

Since the 36 SCoT primers were originally developed for *Oryza sativa* [14] and 60 InDel primers were developed for *C*. × *unshiu* [9], we first assessed the transferability of these primers to 11 citrus accessions. The efficient SCoT and InDel markers were selected for further evaluation (Appendix A).

The SCoT PCR reaction mixture consisted of a 10 μL 2×HS-TaqPCR reaction buffer (Biolabmix, Novosibirsk, Russia) containing Hot Start Taq-Polymerase, 0.4 μL of primer (10 µM), 2 μL of DNA (20 ng μL^−1^), and DEPC-treated water in a total PCR volume of 20 μL. Amplification was carried out in the MiniAmp thermal cycler (Thermo Fisher Scientific, MA, USA) with the following program: primary denaturation for 5 min at 95 °C, annealing for 35 cycles, denaturation at 95 °C for 1 min, annealing at 52 °C for 1 min, elongation at 72 °C for 2 min, and the final elongation at 72 °C for 5 min. The separation of SCoT fragments was performed on a 2% agarose gel for 2.5 h at 90 V in 1 × TAE buffer.

TIe InDel PCR reaction mixture consisted of 10 μL of 2×HS-TaqPCR reaction buffer (Biolabmix, Novosibirsk, Russia) containing Hot Start Taq-Polymerase, 0.4 μL of primer (10 µM), 2 μL of DNA (20 ng μL^−1^), and DEPC-treated water in a total PCR volume of 20 μL. Amplification was carried out in the MiniAmp thermal cycler (Thermo Fisher Scientific, MA, USA) with the following program: primary denaturation 5 min at 95 °C, annealing 35 cycles, denaturation at 95 °C for 30 s, annealing at 55 °C for 30 s, elongation at 72 °C for 30 s, and the final elongation at 72 °C for 2 min. The separation of InDel fragments was performed on a 3% agarose gel for 2.5 h at 90 V in 1 × TAE buffer.

### 4.3. Statistical Analysis

Genetic diversity parameters were calculated for each SCoT and InDel marker in the citrus collection using the software GeneAlex ver. 6.5 (https://biology-assets.anu.edu.au/GenAlEx/Download.html, 10 April 2023) [48,49] and the online resource [50]. The analysis function ‘Matches’ in GeneAlex ver. 6.5 [48,49] was used to identify genotypes with identical allelic patterns within the dataset. One biological replicate (one tree per genotype) and three technical replicates were assayed for each analyzed parameter. The following parameters were assessed: Na—total number of bands, PIC—polymorphism information content, D—discriminating power, and H—genetic diversity. Subsequently, the model-based clustering method was applied using the software STRUCTURE ver. 2.3.4. (Oxford, UK) [51] to verify the genetic structure within the Citrus collection. The parameters included 50,000 burn-in periods and 50,000 Markov Chain Monte Carlo repetitions using the admixture model with correlated allele models. The software STRUCTURE HARVESTER (https://taylor0.biology.ucla.edu/structureHarvester/, 10 April 2023) [52] was used to detect the most likely value for K based on Evanno’s ΔK method [53]. Phylogenetic trees were drawn based on the dissimilarity matrix using DARWIN ver.6.0 [54].

## 5. Conclusions

In our study, the efficiency of SCoT and InDel markers were evaluated in a wide range of citrus species and cultivars including 172 genotypes and 31 species. The highest discriminating power was observed in the following markers: SCoT18 (D = 0.84), SCoT20 (D = 0.93), SCoT23 (D = 0.87), SCoT31 (D = 0.88), SCoT36 (D = 0.87) и LG 1-4 (D = 0.94), LG 4-3 (D = 0.86), LG 7-11 (D = 0.98), and LG 8-10 (D = 0.83), indicating their strong applicability to assess the intraspecific diversity of the genus *Citrus*. The highest levels of genetic admixtures were detected in lime accessions. A low level of intraspecific diversity was detected in satsuma and lemon accessions. However, 10–50% of genetic admixtures were detected in some locally derived accessions, such as satsuma, lemon, sweet orange, and pomelo. These genotypes will be useful for further controlled hybridization to increase intraspecific genetic diversity in the aforementioned species.

Additionally, these results are valuable for further collection management and show the necessity of interspecific hybridization to increase intraspecific genetic diversity in colder regions.

## Figures and Tables

**Figure 1 ijms-24-08276-f001:**
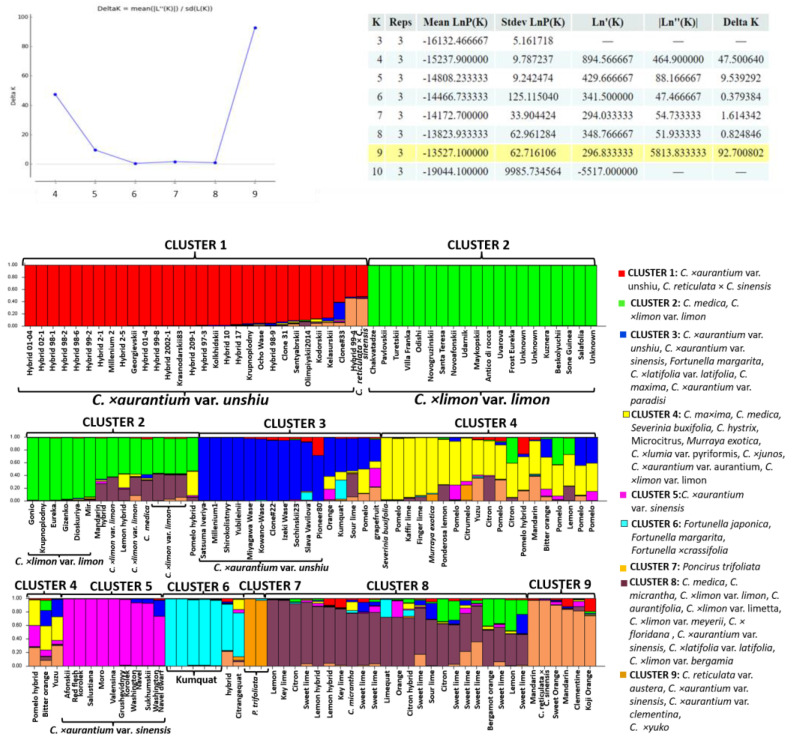
Genetic structure among 172 citrus accessions based on SCoT data. Each colored segment represents the estimated membership fraction of each genetic cluster. The yellow color in the table indicates the most reliable K value (number of genetic clusters) in the collection.

**Figure 2 ijms-24-08276-f002:**
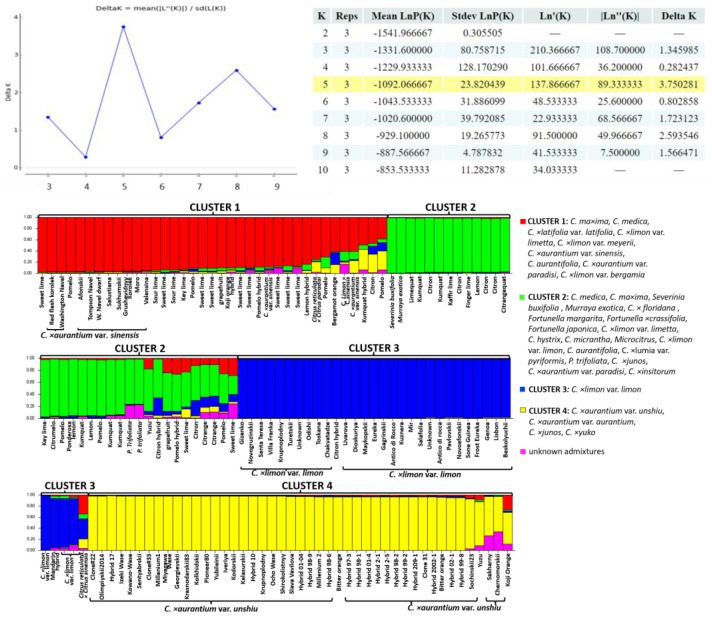
Genetic structure among 172 citrus accessions based on InDel data. Each colored segment represents the estimated membership fraction of each genetic cluster. The yellow color in the table indicates the most reliable K value (number of genetic clusters) in the collection.

**Figure 3 ijms-24-08276-f003:**
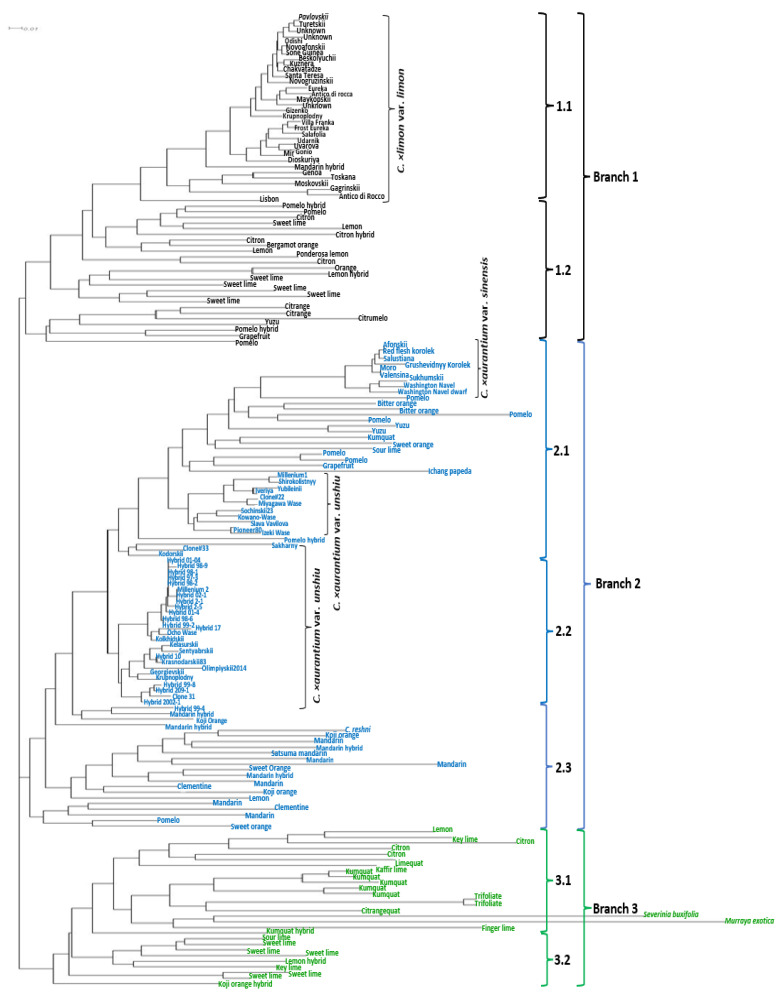
Phylogenetic relationships among 172 citrus genotypes calculated by 12 SCoT and 9 InDel marker data. The numbers on the figure indicate the related branches.

**Table 1 ijms-24-08276-t001:** Genetic diversity parameters of 12 SCoT markers (N = 172).

SCOT	Primers Sequence 5′-3′	Na	H	PIC	D
SCoT2	CAACAATGGCTACCACCC	28	0.33	0.39	0.26
SCoT6	CAACAATGGCTACCACGC	26	0.45	0.34	0.26
SCoT9	CAACAATGGCTACCAGCA	29	0.45	0.34	0.21
SCoT12	ACGACATGGCGACCAACG	35	0.43	0.35	0.28
SCoT13	ACGACATGGCGACCATCG	31	0.46	0.34	0.24
SCoT14	ACGACATGGCGACCACGC	34	0.43	0.35	0.27
SCoT18	ACCATGGCTACCACCGCC	30	0.48	0.32	0.84
SCoT20	ACCATGGCTACCACCGCG	20	0.38	0.37	0.93
SCoT21	ACGACATGGCGACCCACA	28	0.43	0.35	0.29
SCoT23	CACCATGGCTACCACCAG	20	0.46	0.33	0.87
SCoT31	CCATGGCTACCACCGCCT	22	0.45	0.34	0.88
SCoT36	GCAACAATGGCTACCACC	19	0.46	0.33	0.87
MEAN		26.83	0.43	0.34	0.52
SD		5.49	0.04	0.02	0.32

Na—amplicon numbers, PIC—polymorphism information content, D—discriminating power; H—genetic diversity. The background color indicates the minimum (green) and maximum (yellow) values.

**Table 2 ijms-24-08276-t002:** Genetic diversity parameters of 9 InDel markers (N = 172).

InDel	Primers Sequence 5′-3′	Na	H	PIC	D
LG 1-4	F: TACACAGAACCGCCAAATCA R: TCTCCCATGAACCAGCTACC	7	0.36	0.42	0.94
LG 2-6	F: CGCGTGTTACTTCTTGACAGA R: CGAGGCATGTGCTTGAATAA	2	0.22	0.46	0.24
LG3-14	F: TGCCGGGAGTCTTAAAGATG R: CGAGATGGCCACCTAGAAAT	2	0.37	0.42	0.44
LG4-2	F: GGGTTTCTAAGCATTTGGCCA R: ACACTCATCTTCTCGAGCAAAGA	3	0.41	0.41	0.41
LG 4-3	F: AAGAGGACATAAGAGGCAAGTTT R: GCCAAGCAAAACTGATAGGG	4	0.47	0.38	0.86
LG 5-20	F: GGCATTTGAGCTAGAAATTCGT R: AACACTGTCAAAAGAAAACCACA	3	0.44	0.39	0.54
LG 7-11	F: ATTTTGACACGTTCAGCCGC R: TGGATTTTGCACTCACCCTT	11	0.26	0.45	0.98
LG 8-10	F: TCTGCTGACCTTGCTTACGA R: CCCTCACAAGACAGTTGAGGA	4	0.48	0.37	0.83
LG 9-2	F: GGTGATTTTGAGTATGAGAGGTGG R: AGGGTAGTTTTATGATAGTTATCCACA	3	0.41	0.40	0.50
MEAN		4.33	0.38	0.41	0.64
SD		2.92	0.09	0.03	0.27

Na—amplicon numbers, PIC—polymorphism information content, D—discriminating power; H—genetic diversity. The background color indicates the minimum (green) and maximum (yellow) values.

## Data Availability

Not applicable.

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
