# Peer review of "InDel and SCoT Markers for Genetic Diversity Analysis in a Citrus Collection from the Western Caucasus"

_ijms, 2023, doi:10.3390/ijms24098276_

Round 1

Reviewer 1 Report

The journal format should be followed for references. The MS is really well written, nevertheless, in all other respects.

English 

Author Response

Response to Reviewer 1

Point 1: The journal format should be followed for references. The MS is really well written, nevertheless, in all other respects.

Response 1: Dear reviewer, thank you for your time and useful suggestion. we revised references and updated them

Reviewer 2 Report

- Although the abstract already has an introduction, problem statement, statement of a possible solution, selection, and final conclusion, no clear-cut future recommendations, and some quantitative data are provided to attract the interest of the reader. clarify these in 1-2 lines.

- In general, this manuscript is easy to read and understand

 - Different DNA markers like AFLP, IRAP, SSR, CAPS, SCoT, and InDel  were evaluated to characterize the genetic diversity in the plant's germplasm collections, however, authors used the  SCoT markers and InDel markers in this study. Is there a specific reason to use them or ignore other?

- In the introduction part, Ramos et. al. (2006) is rather old to support the authors' claim and talked about the importance of many citrus genotypes that are able to reproduce asexually by apomixis!  Consider updating the reference, if possible. I highly encourage you to cite more up-to-date references.

- The experimental design was not clear, appropriate, reasonable, and sufficiently controlled!

 - On the other hand, the sampling method (sampling time, organ/tissue, mass etc.) for gene expression studies should be described.

- The statistical treatments used should be mentioned.

- I highly encourage you to cite more up-to-date references when referring to agricultural statistics or the current state of farming.

- In the Results section, do not repeat the numbers reported in the Figures in the text. This will hinder readability.

- Conclusion and Future Directions: It can be strengthened by citing specific lacunae and pathways to meet that.

- JUST a few of the references is one or two years old while others are more than ten or fifteen years old! please modify them to newer ones!

-  DOES this kind of study need 78 references? please modify it to less!

- References should follow this journal style!

NEED more!!

Author Response

Response to Reviewer 2

Dear reviewer, thank you for your time and useful suggestions. We have tried to do our best to follow all your advises. However, if we miss something, or if you are still not satisfied we are ready to proceed to improve necessary points according to your requirements.

Point 1: Although the abstract already has an introduction, problem statement, statement of a possible solution, selection, and final conclusion, no clear-cut future recommendations, and some quantitative data are provided to attract the interest of the reader. clarify these in 1-2 lines.

Response 1: We revised the abstract

Point 2: In general, this manuscript is easy to read and understand.

Different DNA markers like AFLP, IRAP, SSR, CAPS, SCoT, and InDel  were evaluated to characterize the genetic diversity in the plant's germplasm collections, however, authors used the  SCoT markers and InDel markers in this study. Is there a specific reason to use them or ignore other?

Response 2: Dear reviewer, thank you for the interest. We used InDel markers because they were recently developed for satsuma mandarins, which usually characterized by the low level of intraspecific diversity, so we suggested that these markers will be extremely efficient. Additionally, SCOT markers are perspective for the QTLs mapping because these multilocus markers are linked with the start-codon ATG, and amplify the exone regions in genome. It is written in the introduction part.

Point 3: In the introduction part, Ramos et. al. (2006) is rather old to support the authors' claim and talked about the importance of many citrus genotypes that are able to reproduce asexually by apomixis!  Consider updating the reference, if possible. I highly encourage you to cite more up-to-date references.

Response 3: We revised references and updated them

Point 4: The experimental design was not clear, appropriate, reasonable, and sufficiently controlled!

On the other hand, the sampling method (sampling time, organ/tissue, mass etc.) for gene expression studies should be described.

Response 4: The details are added to MM part

Point 5: The statistical treatments used should be mentioned.

Response 5: details are added in MM part

Point 6: I highly encourage you to cite more up-to-date references when referring to agricultural statistics or the current state of farming.

Response 6: We revised references and updated them

Point 7: In the Results section, do not repeat the numbers reported in the Figures in the text. This will hinder readability.

Response 7: Revised

Point 8: Conclusion and Future Directions: It can be strengthened by citing specific lacunae and pathways to meet that.

Response 8: Revised accordingly

Point 9: JUST a few of the references is one or two years old while others are more than ten or fifteen years old! please modify them to newer ones!

Response 9: We revised references and updated them

Point 10: DOES this kind of study need 78 references? please modify it to less!

Response 10: Revised

Point 11: References should follow this journal style!

Response 11: Revised

Reviewer 3 Report

To,

The Chief Editor,

IJMS, MDPI,

Manuscript ID: ijms-2368288

Subject: Submission of comments on the manuscript in “IJMS"

Dear Chief Editor IJMS, MDPI,

Thank you very much for the invitation to consider a potential reviewer for the manuscript (ID: ijms-2368288). My comments responses are furnished below as per each reviewer’s comments. 

Dear Chief Editor,

In the reviewed manuscript, the authors the efficiency of 36 SCoT markers and 60 InDel markers were evaluated for the broad Citrus collection of the Western Caucasus. The interspecific and intraspecific genetic diversity and genetic structure were analyzed for 172 accessions including 31 species and sets of the locally derived cultivars. The efficient markers were identified: SСoT18, SсoT20, SсoT23, SсoT31, SсoT36 и LG 1-4, LG 4-3, LG 7-11, LG 8-10. Based on these markers, the genetic admixtures and relationships in the Citrus collection were established. According to STRUCTURE HARVESTER analysis, SCoT and InDel markers showed K = 9 and K = 5 genetic clusters, respectively. The lowest levels of genetic admixtures were observed among the satsuma mandarins and lemons. Overall, SCoT markers showed higher levels of genetic admixtures in citrus collection than InDel markers. Phylogenetic relationships indicated 3 big branches and high level of interspecific genetic diversity. However, low level of intraspecific diversity was observed in locally derived satsuma mandarins. The results provide new insight into the citrus germplasm origin and distribution in the colder regions. However, in my opinion, the MS needs major revisions. I have some suggestions to improve this manuscript: 

  1. The structure of the abstract should be improved, as well as the lack of several aspects that should be included in this section. The abstract should highlight the most important results of the parameters and characteristics assayed.

2.    Introduction part is not impressive and systematic. In the introduction part, the authors should elaborate the scientific issues in the fruit crop research.

3.    Materials and methods must be last section and accoring do renumbering.

  1. The figures are quite low resolution and difficult to make out. Higher-resolution versions will be needed for publication. Further, text in figure is not readble. for example, in Figures 1, 2, and 3.
  2. In Material and Methods:- indicate how many replicates assayed in each analysis/parameter. The number of samples or biological and technical replicates should be mentioned for each parameter in the methods.
  3. Results must be explained clearly and in detail.
  4. Likewise, the discussion part is also very lengthy and complicated. Information should be comprehensively discussed.  
  5. The conclusion part is very week. Improve by adding the results of your studies.
  6. References: shall have to correct the whole References according to the ”Instructions for the Authors”, e.g. title should not be in italics, the Journal name is in italics, and the author shall have to use the abbreviated name Journals cited the year must be bold, the scientific name must be italics etc. Please check all references carefully.

Best wishes and thank you

To,

The Chief Editor,

IJMS, MDPI,

Manuscript ID: ijms-2368288

Subject: Submission of comments on the manuscript in “IJMS"

Dear Chief Editor IJMS, MDPI,

Thank you very much for the invitation to consider a potential reviewer for the manuscript (ID: ijms-2368288). My comments responses are furnished below as per each reviewer’s comments. 

Dear Chief Editor,

In the reviewed manuscript, the authors the efficiency of 36 SCoT markers and 60 InDel markers were evaluated for the broad Citrus collection of the Western Caucasus. The interspecific and intraspecific genetic diversity and genetic structure were analyzed for 172 accessions including 31 species and sets of the locally derived cultivars. The efficient markers were identified: SСoT18, SсoT20, SсoT23, SсoT31, SсoT36 и LG 1-4, LG 4-3, LG 7-11, LG 8-10. Based on these markers, the genetic admixtures and relationships in the Citrus collection were established. According to STRUCTURE HARVESTER analysis, SCoT and InDel markers showed K = 9 and K = 5 genetic clusters, respectively. The lowest levels of genetic admixtures were observed among the satsuma mandarins and lemons. Overall, SCoT markers showed higher levels of genetic admixtures in citrus collection than InDel markers. Phylogenetic relationships indicated 3 big branches and high level of interspecific genetic diversity. However, low level of intraspecific diversity was observed in locally derived satsuma mandarins. The results provide new insight into the citrus germplasm origin and distribution in the colder regions. However, in my opinion, the MS needs major revisions. I have some suggestions to improve this manuscript: 

  1. The structure of the abstract should be improved, as well as the lack of several aspects that should be included in this section. The abstract should highlight the most important results of the parameters and characteristics assayed.

2.    Introduction part is not impressive and systematic. In the introduction part, the authors should elaborate the scientific issues in the fruit crop research.

3.    Materials and methods must be last section and accoring do renumbering.

  1. The figures are quite low resolution and difficult to make out. Higher-resolution versions will be needed for publication. Further, text in figure is not readble. for example, in Figures 1, 2, and 3.
  2. In Material and Methods:- indicate how many replicates assayed in each analysis/parameter. The number of samples or biological and technical replicates should be mentioned for each parameter in the methods.
  3. Results must be explained clearly and in detail.
  4. Likewise, the discussion part is also very lengthy and complicated. Information should be comprehensively discussed.  
  5. The conclusion part is very week. Improve by adding the results of your studies.
  6. References: shall have to correct the whole References according to the ”Instructions for the Authors”, e.g. title should not be in italics, the Journal name is in italics, and the author shall have to use the abbreviated name Journals cited the year must be bold, the scientific name must be italics etc. Please check all references carefully.

Best wishes and thank you

Author Response

Response to Reviewer 3

Dear reviewer, thank you for your time and useful suggestions. We really appreciate this. We have tried to do our best to follow all your advises. However, if we miss something, or if you are still not satisfied, we are ready to proceed to improve necessary points according to your requirements.

Point 1: The structure of the abstract should be improved, as well as the lack of several aspects that should be included in this section. The abstract should highlight the most important results of the parameters and characteristics assayed.

Response 1: We have tried to revise the abstract accordingly.

Point 2: Introduction part is not impressive and systematic. In the introduction part, the authors should elaborate the scientific issues in the fruit crop research.

Response 2:The introduction part is revised. We have tried to make more emphasis on the scientific issues.

Point 3: Materials and methods must be last section and according do renumbering.

Response 3: We revised this part accordingly

Point 4: The figures are quite low resolution and difficult to make out. Higher-resolution versions will be needed for publication. Further, text in figure is not readble. for example, in Figures 1, 2, and 3.

Response 4: We improved the quality of figures

Point 5: In Material and Methods:- indicate how many replicates assayed in each analysis/parameter. The number of samples or biological and technical replicates should be mentioned for each parameter in the methods.

Response 5: We have added the details to MM part section Statistical analysis.

Point 6: Results must be explained clearly and in detail.

Response 6: We revised this part to make it more clear.

Point 7: Likewise, the discussion part is also very lengthy and complicated. Information should be comprehensively discussed.  

Response 7: We revised discussion according to your suggestion.

Point 8: The conclusion part is very week. Improve by adding the results of your studies.

Response 8: We revised conclusion part to make it more clear.

Point 9: References: shall have to correct the whole References according to the ”Instructions for the Authors”, e.g. title should not be in italics, the Journal name is in italics, and the author shall have to use the abbreviated name Journals cited the year must be bold, the scientific name must be italics etc. Please check all references carefully.

 Response 9: We have revised the references.

Round 2

Reviewer 3 Report

Dear Chief Editor,

Thank you for providing the opportunity to review the revised manuscript. The manuscript is improved considerably after revision according to the reviewer's comment. Now this study is a suitable contribution to the IJMS. I recommend the manuscript for publication.

Thank you

With best regards

Dear Chief Editor,

Thank you for providing the opportunity to review the revised manuscript. The manuscript is improved considerably after revision according to the reviewer's comment. Now this study is a suitable contribution to the IJMS. I recommend the manuscript for publication.

Thank you

With best regards